# A generalized density-modulated twist-splay-bend phase of banana-shaped particles

Massimiliano Chiappini🄳 [1✉] & Marjolein Dijkstra🄳 [1✉]

In 1976, Meyer predicted that bend distortions of the nematic director field are complemented by deformations of either twist or splay, yielding twist-bend and splay-bend nematic phases, respectively. Four decades later, the existence of the splay-bend nematic phase remains dubious, and the origin of these spontaneous distortions uncertain. Here, we conjecture that bend deformations of the nematic director can be complemented by simultaneous distortions of both twist and splay, yielding a twist-splay-bend nematic phase. Using theory and simulations, we show that the coupling between polar order and bend deformations drives the formation of modulated phases in systems of curved rods. We find that twist-bend phases transition to splay-bend phases via intermediate twist-splay-bend phases, and that splay distortions are always accompanied by periodic density modulations due to the coupling of the particle curvature with the non-uniform curvature of the splayed director field, implying that the twist-splay-bend and splay-bend phases of banana-shaped particles are actually smectic phases.

[1] Soft Condensed Matter, Debye Institute for Nanomaterials Science, Department of Physics, Utrecht University, Utrecht, The Netherlands. ✉email: m.chiappini@uu.nl; m.dijkstra@uu.nl

The simplest and most common liquid crystal phase is the nematic phase, which is also the most relevant one for optoelectronic applications. The uniaxial nematic (N) phase consists of anisotropic particles that lack positional order but display orientational order as the particles are preferentially aligned along a so-called nematic director $\hat{\mathbf{n}}$. More exotic states of matter can be conjectured if the nematic director is allowed to vary in space, i.e. the average orientation of a particle at position $\mathbf{r}$ is defined by a nematic director field $\hat{\mathbf{n}}(\mathbf{r})$. A well-known example is the chiral nematic or cholesteric (N*) phase, which is characterised by a nematic director that rotates as a helix around a chiral director with a cholesteric pitch length $p$, denoting the length scale associated to the helical periodicity.

An even more fascinating example is the twist-bend nematic ($N_{TB}$) phase, recently discovered in experiments on bent-core mesogens[1–12]. The $N_{TB}$ phase was already predicted by Meyer in 1976[13] and by Dozov in 2001[14] for banana-shaped particles that favor spontaneous bend deformations in the nematic director field. As a pure bend deformation cannot uniformly fill the three-dimensional space, local bend deformations have to be accompanied by either a spontaneous twist, yielding an $N_{TB}$ phase, or by splay distortions, resulting into an oscillating splay-bend nematic ($N_{SB}$) phase, see Fig. 1a, c. The $N_{TB}$ phase is characterised by a nematic director field that precesses around a right circular cone with a pitch $p$ and a conical angle $0 < \theta_0 < \pi/2$. Hence, the $N_{TB}$ phase is a chiral phase with local polar order and a uniform bend deformation. Because of the achirality of bent-core mesogens, the precession of the nematic director of the $N_{TB}$ phase can be left- or right-handed. On the other hand, the nematic director field of the $N_{SB}$ phase precesses over a flat isosceles triangle with maximum angle $\theta_0$, thereby preserving the achiral symmetry and oscillating between non-uniform bend and splay domains, see Supplementary Note 2.

Many fundamental questions regarding the $N_{TB}$ phase are still open despite numerous theoretical and experimental investigations. Even the most basic question regarding the origin of bend deformations and the appearance of chiral symmetry in systems of achiral bent-shaped particles is still unresolved. While Meyer invoked that bend deformations originate from the spontaneous polar ordering of the particles due to either the molecular shape or the electrostatic polarization[13,15], Dozov ignored the possibility of spontaneous polar order and explained the bend distortions by a negative bend elastic constant[14]. In addition, the relationship between the structure of the $N_{TB}$ phase and the details of the constituent molecules is still not well understood. It is found experimentally that the observation of the $N_{TB}$ phase depends sensitively on the molecular details.

Flexible bent-core molecules linked with an odd number of hydrocarbon atoms display $N_{TB}$ phases but not the ones with an even-numbered linkage[16]. On the other hand, computer simulations demonstrate the existence of $N_{TB}$ phases for both rigid and flexible banana-shaped molecules[17–20]. Flexibility also plays an important role in the stabilisation of $N_{TB}$ phases as most rigid bent-core molecules form smectic (Sm) phases instead of nematic phases. Finally, the prediction of the surprisingly short pitch length and of the non-trivial tilt angle of the helicoidal nematic director field on the basis of the microscopic details of the particles is also of urgent interest for the design of optoelectronic materials.

On the other hand, even though the Oseen–Frank theory predicts the $N_{SB}$ to be more stable than the $N_{TB}$ phase when the splay elastic constant is smaller than twice the twist elastic constant, experimental evidence of a $N_{SB}$ phase is still lacking. Recent simulations suggest that an $N_{SB}$ phase could be stabilised in lyotropic systems of banana-shaped particles[20] but the presence of long-ranged density modulations questions the nematic nature of this $N_{SB}$ phase. We also note that a transition from an $N_{TB}$ phase to an unidentified density-modulated ($Sm_X$) phase was recently observed in experiments[21]. Yet it is unclear what the physical mechanism is behind the $N_{SB}$ phase and how the system transforms into an $N_{SB}$ phase. Does the transformation from the N or $N_{TB}$ phase proceed via a first-order or second-order phase transition? Here we conjecture that the transition from the $N_{TB}$ to the $N_{SB}$ phase may proceed via an intermediate phase that

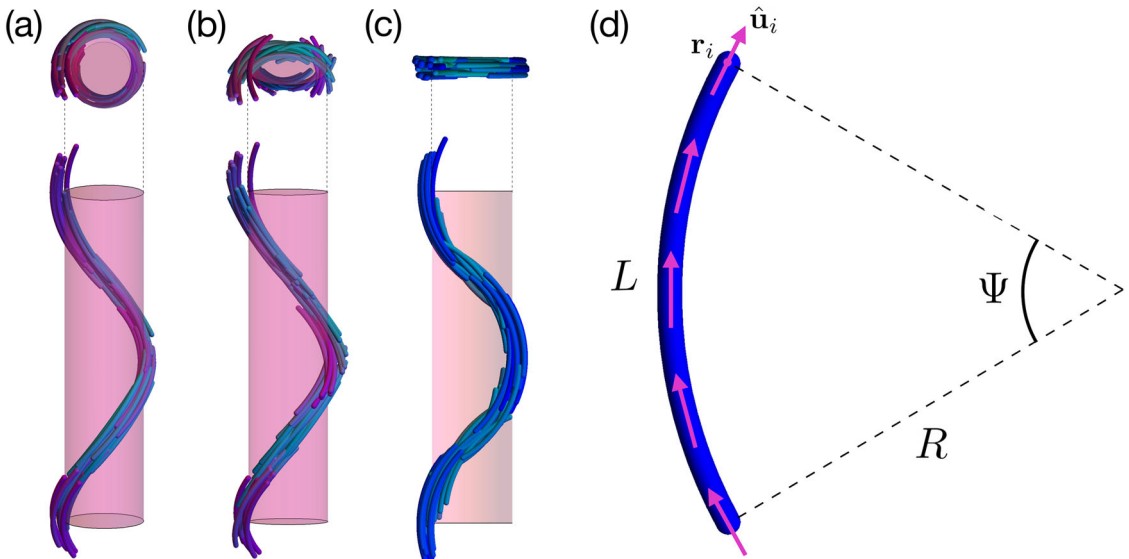

**Fig. 1 Schematics of a twist-bend, twist-splay-bend, and splay-bend nematic phase of curved rods. a–c** Side and top views of the spatial modulations of the particle orientations in (**a**) a chiral twist-bend nematic ($N_{TB}$) phase, **b** a chiral twist-splay-bend nematic ($N_{TSB}$) phase and (**c**) an achiral splay-bend nematic ($N_{SB}$) phase. **d** A hard curved spherocylinder consisting of a cylinder of length $L$ and diameter $D$ capped at both ends with a hemisphere of diameter $D$ and bent with a radius of curvature $R$ corresponding to an opening angle $\Psi = L/R$. In our generalized Maier–Saupe theory, this particle is modelled as a rigid chain of $M$ segments with center-of-mass positions $\mathbf{r}_i$ and orientations $\hat{\mathbf{u}}_i$ tangent to the particle profile for $i \in 1, \cdots, M$, sketched by the (pink) arrows pointing upwards along the arc.

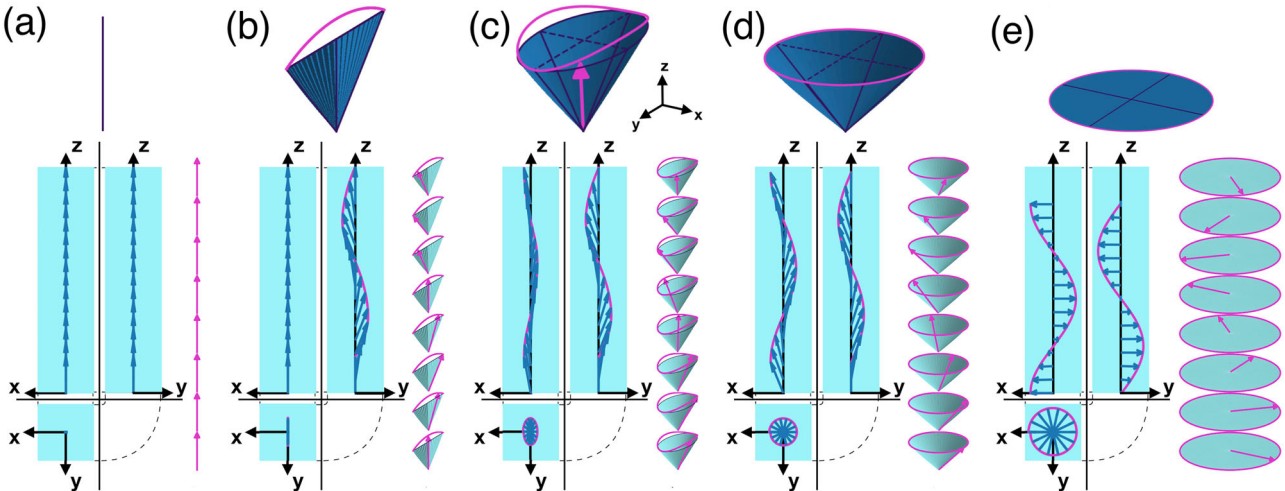

**Fig. 2 Nematic director field of a twist-splay-bend nematic phase.** Precession cone (top panel) and orthogonal projection of the nematic director field $\hat{\mathbf{n}}_{TSB}(z|\theta_a, \theta_b, q)$ (bottom panel) of several limiting cases of the twist-splay-bend nematic ($N_{TSB}$) phase: **a** the uniaxial nematic (N) phase with $\theta_a = \theta_b = 0$, **b** the splay-bend nematic ($N_{SB}$) phase with $\theta_b = 0$, **c** the $N_{TSB}$ phase with $\theta_a \neq \theta_b$, **d** the twist-bend ($N_{TB}$) phase with $\theta_a = \theta_b$, and (**e**) the cholesteric (N*) phase with $\theta_a = \theta_b = \pi/2$.

displays spatial modulations of both twist, splay, and bend, thereby extending Meyer's speculations of 1976 by conjecturing that spontaneous bend deformations of the nematic director field can be accompanied by simultaneous deformations of both twist and splay. In this picture, twist deformations in the $N_{TB}$ phase are gradually replaced by splay deformations, eventually resulting into an $N_{SB}$ phase with pure splay and bend deformations. A similar scenario was recently discovered by studying the response of an $N_{TB}$ phase to an external field[22,23], which undergoes a structural change via an elliptical analogue of the $N_{TB}$ phase to an $N_{SB}$ phase upon increasing the field strength. This intermediate phase that we term the twist-splay-bend nematic ($N_{TSB}$) phase is characterized by a nematic director that precesses around an elliptical cone with conical angles $\theta_a$ and $\theta_b$, see Fig. 2.

To shed light on the microscopic origin of the spatially modulated nematic phases and to better understand the transformation from the $N_{TB}$ to an $N_{SB}$ phase via a possible $N_{TSB}$ phase, we develop a Maier–Saupe-like mean-field theory that takes into account not only the particle shape and interactions, but also the spatial modulations of the nematic director and density field in a variational fashion. We map out a theoretical phase diagram of curved spherocylinders that displays stable $N_{TB}$, twist-splay-bend, and splay-bend phases, and test the predictions against simulations. We show that the twist-splay-bend and splay-bend (smectic) phases present periodic density modulations due to non-uniform deformations in the director field. Finally, we derive a relation between the particle curvature and the structure of the $N_{TB}$ phase.

## Results

### A variational mean-field theory for spatially modulated liquid crystal phases.
We generalize a recent Maier–Saupe theory for thermotropic bent-core mesogens[24] to determine the phase behavior of curved spherocylinders with diameter $D$, length $L$, and radius of curvature $R$ corresponding to a central angle $\Psi = L/R$ (Fig. 1d). We describe a curved spherocylinder with centre-of-mass position $\mathbf{R} = (X, Y, Z)$ and orientation $\Omega = (\alpha, \beta, \gamma)$ as a rigid chain of $M$ segments of length $L/M$. We find that $M = 10$ segments is sufficient to account for the particle shape for the range of $\Psi$ that we considered. Each segment $i$, with centre-of-mass position $\mathbf{r}_i$ and orientation $\hat{\mathbf{u}}_i$, is assumed to align preferentially along the local nematic director $\hat{\mathbf{n}}(\mathbf{r}_i)$ via an effective

mean-field potential $\beta U(\mathbf{R}, \Omega)$. In addition, we employ McMillan's extension[25] to account for possible density modulations along the (global) nematic director. The resulting effective mean-field potential reads

$$\beta U(\mathbf{R}, \Omega) = -\frac{\beta\epsilon}{M}\left[S + \alpha\tau\cos\left(\frac{2\pi Z}{\lambda}\right)\right] \times \left(\sum_{i=1}^{M} P_2(\hat{\mathbf{u}}_i \cdot \hat{\mathbf{n}}(\mathbf{r}_i))\right),$$

(1)

where, $\beta\epsilon$ is a dimensionless constant that quantifies the alignment strength, $P_2(x) = (3x^2 - 1)/2$ is the second-order Legendre polynomial, $\beta = 1/k_BT$ is the inverse temperature with $k_B$ Boltzmann's constant, $\lambda$ is the periodicity of the density modulations, and $\alpha$ is a tunable parameter determining the tendency of the system to form density modulations. Furthermore, $S$ and $\tau$ are the local nematic and the smectic order parameters, respectively,

$$S = \left\langle \frac{1}{M}\sum_{i=1}^{M} P_2(\hat{\mathbf{u}}_i \cdot \hat{\mathbf{n}}(\mathbf{r}_i)) \right\rangle, \text{ and}$$

$$\tau = \left\langle \cos\left(\frac{2\pi Z}{\lambda}\right)\left(\frac{1}{M}\sum_{i=1}^{M} P_2(\hat{\mathbf{u}}_i \cdot \hat{\mathbf{n}}(\mathbf{r}_i))\right) \right\rangle,$$

(2)

where $\langle \cdots \rangle = \int d\mathbf{R}d\Omega f(\mathbf{R}, \Omega) \cdots$ denotes the ensemble average with the probability distribution function $f(\mathbf{R}, \Omega) = \exp[-\beta U(\mathbf{R}, \Omega)]/Q$ for the position and orientation of a curved spherocylinder, and $Q = \int d\mathbf{R} d\Omega \exp[-\beta U(\mathbf{R}, \Omega)]$ is the partition function.

As a variational ansatz for $\hat{\mathbf{n}}(\mathbf{r})$ we employ the nematic director field $\hat{\mathbf{n}}_{TSB}(z|\theta_a, \theta_b, q)$ of a generic $N_{TSB}$ phase with conical angle $\theta_a$ and $\theta_b$, wavenumber $q$, and the helical axis along the $z-$direction

$$\hat{\mathbf{n}}_{TSB}(z|\theta_a, \theta_b, q) = \sin(\theta_b)\cos(qz)\,\mathbf{i} + \sin(\theta_a)\sin(qz)\,\mathbf{j}$$
$$+ \sqrt{1 - \sin^2(\theta_b)\cos^2(qz) - \sin^2(\theta_a)\sin^2(qz)}\,\mathbf{k}.$$

(3)

We note that the $N_{TSB}$ phase reduces to an $N_{TB}$ phase with a circular precession cone when $\theta_a = \theta_b = \theta_0$, whereas for $\theta_a = \theta_b = \pi/2$ the circular cone reduces to a flat circle resulting into an N* phase as the precession of the nematic director reduces to a simple twist around the phase director. If either $\theta_a$ or $\theta_b$ vanishes, the elliptical cone collapses onto a flat isosceles triangle and an $N_{SB}$ phase is obtained with nematic director field

$\hat{\mathbf{n}}_{\mathrm{TSB}}(z|\theta_0, 0, q) = \sin(\theta_0)\sin(qz)\,\mathbf{j} + \sqrt{1 - \sin^2(\theta_0)\sin^2(qz)}\,\mathbf{k}$ which differs from the expression $\hat{\mathbf{n}}_{\mathrm{SB}}(z) = \sin(\theta_0 \sin(qz))\,\mathbf{j} + \cos(\theta_0 \sin(qz))\,\mathbf{k}$ given by Dozov[14]. However, for small conical angles the two expressions for the nematic director fields are indistinguishable, see Supplementary Note 1. Finally, for $\theta_a = \theta_b = 0$ the cone becomes a simple line, and the $N_{\mathrm{TSB}}$ simply reduces to a uniaxial N phase. Hence, all the above-mentioned nematic phases are limiting cases of the $N_{\mathrm{TSB}}$ phase, see Fig. 2. In the following, we distinguish $N_{\mathrm{TB}}$, $N_{\mathrm{TSB}}$, and $N_{\mathrm{SB}}$ states based on their ellipticity $e = \theta_b/\theta_a$: we label $N_{\mathrm{TB}}$ the states with $e > 0.8$, $N_{\mathrm{TSB}}$ the states with $0.2 < e < 0.8$, and $N_{\mathrm{SB}}$ the states with $e < 0.2$. We note that the thresholds $e = 0.2$ and $e = 0.8$ — necessary because of the statistical error on $e$ — are arbitrary, and every state with ellipticity $e \neq 0, 1$ is, in principle, an $N_{\mathrm{TSB}}$ state.

As $\beta U(\mathbf{R}, \Omega)$ only depends on the $Z$-component of $\mathbf{R}$ with period $p$, we restrict all integrations over $\mathbf{R}$ to integrations over $Z \in [0, p]$. The onset of orientational and/or positional order corresponds to a change of entropy per particle $\Delta S/N = -k_B \int dZ d\Omega\, f(Z, \Omega) \log[16\pi^3 f(Z, \Omega)/q] = \langle U(Z, \Omega)\rangle/T - k_B \log(16\pi^3/qQ)$ and to a change of energy per particle $\Delta U/N = \langle U(Z, \Omega)\rangle/2$. Using Eq. (2), we obtain the change of free energy per particle relative to the isotropic state

$$\frac{\beta \Delta F}{N} = \frac{\beta \Delta U}{N} - \frac{\Delta S}{k_B N} = \beta \epsilon \left( \frac{S^2 + \alpha \tau^2}{2} \right) - \log\left( \frac{qQ}{16\pi^3} \right). \quad (4)$$

Minimizing $\Delta F$ with respect to $S$, $\tau$, $\lambda$, and the variational parameters $\theta_a$, $\theta_b$, and $q$ of the nematic director field $\hat{\mathbf{n}}_{\mathrm{TSB}}(z|\theta_a, \theta_b, q)$ yields the equilibrium phase at temperature $k_B T/\epsilon$. We note that since the nematic director $\hat{\mathbf{n}}_{\mathrm{TSB}}(z|\theta_a, \theta_b, q)$ already imposes a periodicity with pitch length $p = 2\pi/q$ in the system, $p$ must be a multiple of the periodicity $\lambda$ of the density modulations, *i.e.* $p = n\lambda$ with $n \in \mathbb{N}$. As the numerical minimization of $\Delta F$ always yields non-integer values of $n$ close to 2 in spatially modulated phases, we impose $n = 2$ in all considered cases.

In Fig. 3a we show the resulting phase diagram from Maier–Saupe theory for a system of curved spherocylinders as a function of $\Psi$ and inverse temperature $\beta \epsilon$, where we set $\alpha = 0.05$. At low curvature, *i.e.* small $\Psi$, the I phase transforms into a uniaxial N phase and subsequently into an $N_{\mathrm{TB}}$ phase upon increasing $\beta \epsilon$. However, the stability range of the N phase shrinks with increasing particle curvature, eventually disappearing at $\Psi \gtrsim 1.2$. Our results show that deformations of the nematic director field become more pronounced with increasing particle curvature $\Psi$ until the I-N phase transition is replaced by a direct I-$N_{\mathrm{TB}}$ transition[20,24,26].

Moreover, our generalized Maier–Saupe theory predicts that upon increasing $\beta \epsilon$ further, the nematic director field exhibits not only twist and bend modulations but also splay deformations, resulting in a twist-splay-bend phase. Upon lowering the temperature further, the twist deformations are gradually replaced by splay deformations, eventually yielding a splay-bend phase at sufficiently high $\beta \epsilon$. In particular, we find two distinct regions of splay-bend phases, one at low particle curvature and one at high curvature, the latter transforming into a re-entrant twist-splay-bend phase with increasing $\beta \epsilon$. Intriguingly, our theory predicts that the onset of splay deformations is accompanied by density modulations. We therefore refer to these phases as twist-splay-bend smectic (Sm$_{\mathrm{TSB}}$) and splay-bend smectic (Sm$_{\mathrm{SB}}$) phases rather than $N_{\mathrm{TSB}}$ and $N_{\mathrm{SB}}$ phases. This finding is also supported by the observation that in the case of $\alpha = 0$, i.e. without McMillan's extension to describe density modulations, the phase diagram displays only stable I, N, and $N_{\mathrm{TB}}$ phases. The $N_{\mathrm{SB}}$ phase is thus unstable in the case of a homogeneous density field. Furthermore, we find that varying the value of $\alpha > 0$ results only in a temperature shift of the transition from $N_{\mathrm{TB}}$ to the Sm$_{\mathrm{TSB}}$ and Sm$_{\mathrm{SB}}$ phase, whereas the amplitude of the density modulations remains unaffected. This confirms that splay deformations of the nematic director field are inherently associated with the appearance of density modulations, as already speculated by De Gennes and Meyer, four decades ago[27,28].

**Monte Carlo simulations of hard curved spherocylinders in bulk and sedimentation.** To test the predictions of our Maier–Saupe theory, we study the bulk phase behavior of hard curved spherocylinders with $L/D = 19$ and varying particle curvature $\Psi$ using $NPT$-MC simulations, i.e. the number of particles $N$, pressure $P$ and temperature $T$ are fixed. The resulting phase diagram is shown in Fig. 3b as a function of $\Psi$ and packing fraction $\eta$. The phase behavior of this athermal lyotropic system is driven by $\eta$ that plays a similar role as the inverse temperature $\beta \epsilon$ in our Maier–Saupe theory for thermotropic systems. Using this

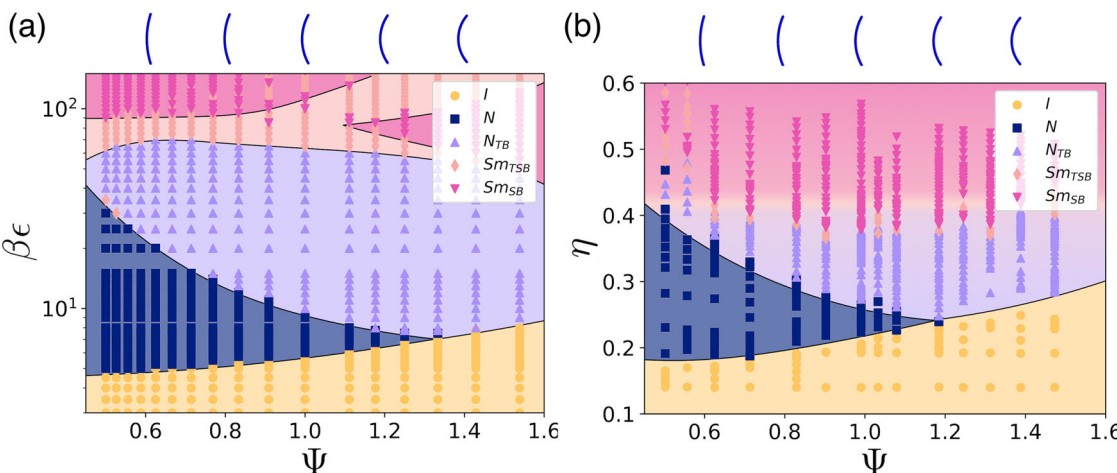

**Fig. 3 Phase diagram of curved spherocylinders.** Phase behaviour of hard curved spherocylinders as (**a**) predicted by theory, as a function of inverse temperature $\beta \epsilon$ and particle curvature $\Psi$, and (**b**) obtained from simulations, as a function of packing fraction $\eta$ and $\Psi$. Both phase diagrams exhibit an isotropic (I) phase at low $\beta \epsilon$ and $\eta$, respectively. For small particle curvatures $\Psi$ the I phase transitions into an uniaxial nematic (N) phase upon increasing $\beta \epsilon$ or $\eta$. The stability range of the N phase decreases with $\Psi$ and disappears at $\Psi \gtrsim 1.2$. Upon increasing $\beta \epsilon$ or $\eta$ further, the twist-bend nematic ($N_{\mathrm{TB}}$) phase transforms into a twist-splay-bend smectic (Sm$_{\mathrm{TSB}}$) phase, and eventually into a splay-bend smectic (Sm$_{\mathrm{SB}}$) phase. The splay deformations in the nematic director field are accompanied by density modulations. Source data are provided as a Source Data file.

analogy, the comparison of the topology of the theoretical with the computational phase diagram is remarkable. Our simulations confirm the I-N-$N_{TB}$ phase sequence at low particle curvature that transforms into $Sm_{TSB}$ and $Sm_{SB}$ phases upon increasing $\eta$, as well as a direct I-$N_{TB}$ transition at high particle curvature transforming into $Sm_{TSB}$ and $Sm_{SB}$ phases with increasing density. Our simulations reveal that the splay modulations are accompanied by density modulations in agreement with our predictions from Maier–Saupe theory. In the Supplementary Note 4, we present a mapping of the theoretical and computational phase diagrams using the dependence of the global nematic order parameter $S_g$ on temperature and packing fraction, and we provide a comparison of the orientational order parameters, pitch and conical angles as a function of thermodynamic state, showing quantitative agreement between the predictions of the Maier–Saupe theory and simulations. We note that the main qualitative difference is the absence of the re-entrant $Sm_{TSB}$ phase in the simulated phase diagram. It is important to mention here that due to hysteresis effects and slow equilibration it is impossible to accurately determine the regions of stability and the first- or second-order nature of the phase transitions of the $N_{TB}$, $Sm_{TSB}$, and $Sm_{SB}$ phases in simulations of hard curved spherocylinders. However, the simulations consistently show that the phase transformation from an $N_{TB}$ to an $Sm_{SB}$ phase proceeds via an intermediate $Sm_{TSB}$ phase as twist deformations are gradually replaced by splay deformations of the nematic director field. For example, in Fig. 4 we show typical configurations of an $Sm_{SB}$ phase, an $Sm_{TSB}$ phase, and an $N_{TB}$ phase along an expansion of a system of hard curved spherocylinders of length $L/D = 19$ and opening angle $\Psi = 1.31$ from packing fraction $\eta = 0.406$ to packing fraction $\eta = 0.367$. To fully characterise the phases in Fig. 4, we plot the scalar order parameter $S(z)$, the nematic director field $\hat{\mathbf{n}}(z)$, and the density field $\rho(z)$ in Supplementary Fig. 12, 13 and 14. In addition, we also plot the polarisation vector $\hat{\mathbf{m}}(z)$ along with the bend distortions in the director field described by the bend vector $\hat{\mathbf{b}}(z) = \hat{\mathbf{n}} \times (\nabla \times \hat{\mathbf{n}})/\| \hat{\mathbf{n}} \times (\nabla \times \hat{\mathbf{n}}) \|$. We clearly observe that $\hat{\mathbf{m}}(z)$ is always anti-parallel to $\hat{\mathbf{b}}(z)$, demonstrating that these modulated phases are driven by bend deformations coupled to polar ordering spontaneously arising from the packing constraints of banana-shaped particles.

Finally, we perform simulations on a system of hard curved spherocylinders with $L/D = 19$ and $\Psi = 0.99$ under gravity with a gravitational length $l_g = k_B T/mg = 7.5D$ parallel to $\hat{\mathbf{z}}$ with a hard wall at the bottom at $z = 0$, $m$ denoting the buoyancy mass of the rods and $g$ the gravitational acceleration. The resulting configuration, presented in Fig. 5a, shows the full I-N-$N_{TB}$-$Sm_{TSB}$-$Sm_{SB}$ phase sequence in a single system. In particular, we observe a continuous transition from an $N_{TB}$ phase with a director field precessing on a circular cone at the top of the sediment, via an $Sm_{TSB}$ phase where the precession cone becomes elliptic, towards an $Sm_{SB}$ phase where the elliptical cone reduces to a flat triangle at the bottom of the sample. Hence, the transition from the $N_{TB}$ to the $Sm_{SB}$ phase occurs via a continuous flattening of the precession cone of the nematic director field from a right circular cone to an isosceles triangle via a continuous range of elliptical precession cones corresponding to a range of $Sm_{TSB}$ phases. In the Supplementary Note 6, we compare the equation of state obtained by integrating the density profile of the sediment with the equation of state obtained from bulk simulations, showing good agreement.

**Rationalising the topology of modulated liquid crystal phases.** Our extensive simulations show that our generalized Maier–Saupe theory effectively predicts, despite its simplicity, the stability of twist-bend, twist-splay-bend, and splay-bend phases of

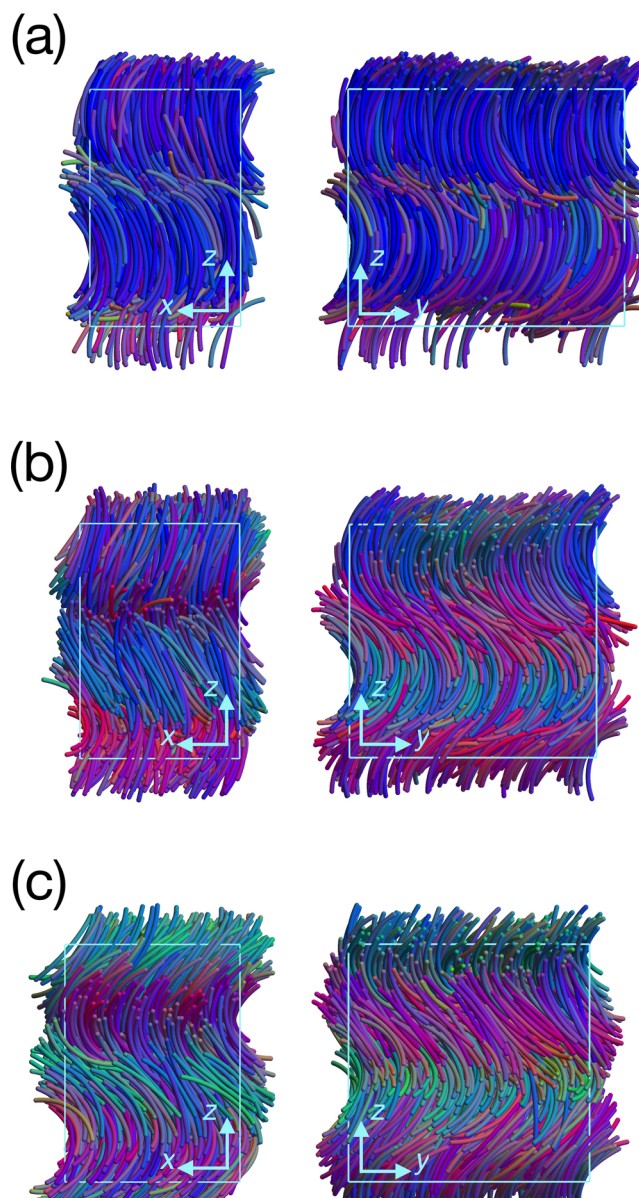

**Fig. 4 Splay-bend smectic $Sm_{SB}$, twist-splay-bend smectic $Sm_{TSB}$, and twist-bend nematic $N_{TB}$ phases.** Typical configurations of (**a**) an $Sm_{SB}$ phase with conical angles $\theta_a \sim 0.76$ and $\theta_b \sim 0$ at packing fraction $\eta = 0.394$ (box size $55.5D \times 30.3D \times 47.7D$), **b** an $Sm_{TSB}$ phase with conical angles $\theta_a \sim 0.87$ and $\theta_b \sim 0.51$ at packing fraction $\eta = 0.371$ (box size $52.0D \times 33.4D \times 49.1D$), and (**c**) an $N_{TB}$ phase with conical angles $\theta_a \sim \theta_b \sim 0.83$ at packing fraction $\eta = 0.354$ (box size $49.9D. 3D \times 49.3D$) along an expansion of a system of hard curved spherocylinders of length $L/D = 19$ and opening angle $\Psi = 1.31$.

curved spherocylinders. Our microscopic theory is solely based on the tendency of particles to align their particle shape to the local nematic director field $\hat{\mathbf{n}}(\mathbf{r})$. To rationalize our findings we calculate the integral curves of the nematic director field $\hat{\mathbf{n}}(z)$, *i.e.* curves $\mathbf{r}(z)$ to which the nematic director $\hat{\mathbf{n}}(z)$ at $z$ is tangent, such that $\mathbf{r}'(z) = \nu \hat{\mathbf{n}}(z)$ with $\nu$ a proportionality constant. We thus find

$$\mathbf{r}(z) = \mathbf{r}(z_0) + \int_{z_0}^{z} d z' \nu \hat{\mathbf{n}}(z'), \quad (5)$$

where the integration constant $\mathbf{r}(z_0) = (x_0, y_0, z_0)$ corresponds to

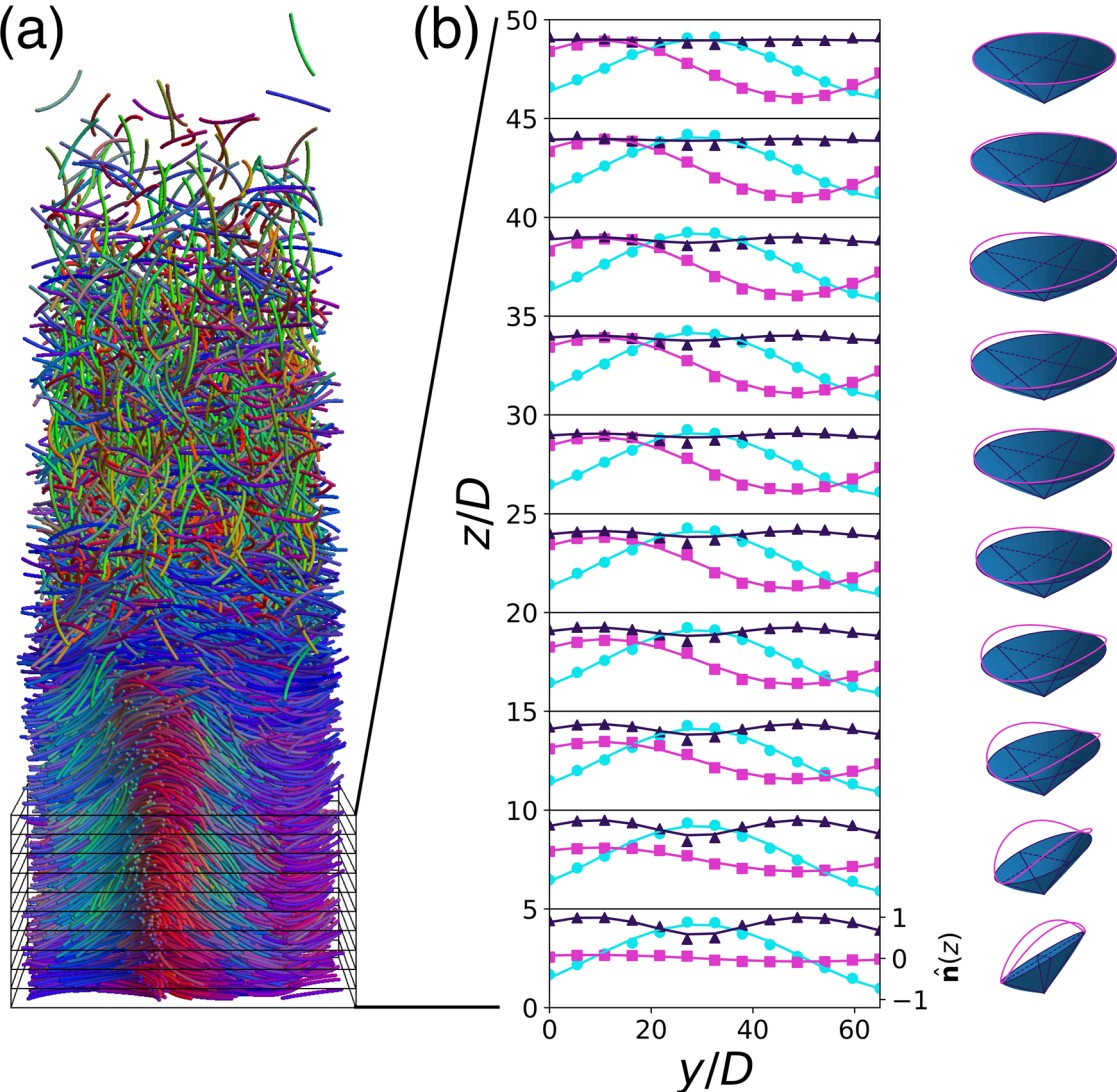

**Fig. 5 A system of curved rods in gravity. a** Typical configuration from a simulation on a system of hard curved spherocylinders with an aspect ratio $L/D$ = 19 and central angle $\Psi = 0.99$ in a gravitational field along $-\hat{z}$ with a gravitational length $l_g = 7.5D$ and a hard wall at $z = 0$. The phase sequence I-N-$N_{TB}$-$Sm_{TSB}$-$Sm_{SB}$ is observed from the top to the bottom in the sediment. **b** The $x$-, $y$-, and $z$-components of the nematic director field $\hat{\mathbf{n}}(y) = (n_x(y), n_y(y), n_z(y))$ in subsequent slabs of the system of thickness $5D$, showing a continuous transition from splay-bend deformations to twist-bend deformations with increasing altitude $z$, corresponding to lower pressure and density. The measured values of the nematic director field are denoted by the varying symbols, whereas the lines denote a fit of the simulation data using Eq. (3). The precession cones correspond to the fits of the nematic director field, showing a continuous transition from an $N_{TB}$ phase with a director field precessing around a circular cone at the top of the sediment, via an $Sm_{TSB}$ phase where the precession cone becomes elliptic, towards an $Sm_{SB}$ phase where the elliptical cone reduces to a flat triangle at the bottom of the sample. Source data for (**b**) are provided as a Source Data file.

the starting point of an integral curve, such that Eq. (5) actually yields an infinite collection of curves, all identical except for a translation in $x$ and $y$.

For a generic twist-splay-bend (TSB) phase the integral in Eq. (5) cannot be evaluated analytically. However, the tendency of particles to align their profiles to the nematic director field at low temperatures or high densities, corresponds to a tendency to match their particle curvature with the curvature of its integral curves. Given a curve $\mathbf{r}(z)$ with tangent $\mathbf{r}'(z) = \nu \hat{\mathbf{n}}(z)$ of constant norm $\parallel \mathbf{r}'(z) \parallel = \nu$, we can reparametrize it by its arc length as $\mathbf{r}(s = \nu z)$. In this reparametrization the tangent to the curve $\mathbf{r}'(s) = (\partial \mathbf{r}(z)/\partial z)z'(s) = \hat{\mathbf{n}}(s)$ has unit norm, and we can calculate the curvature of the integral curve as $\kappa(s) = \parallel \hat{\mathbf{n}}'(s) \parallel$. Going back to the original parametrization, we obtain $\kappa(z) = \parallel \hat{\mathbf{n}}'(z) \parallel / \nu$. Using this formula, we obtain the following curvature for the integral

curve of the nematic director field of a generic TSB phase,

$$\kappa_{\text{TSB}}(z) = \frac{q}{\nu} \sqrt{\frac{\sin^2(\theta_a) - 2\sin^2(\theta_a)\sin^2(\theta_b) + \sin^2(\theta_b) + (\sin^2(\theta_a) - \sin^2(\theta_b))\cos(2qz)}{2(1 - \sin^2(\theta_a)\sin^2(qz) - \sin^2(\theta_b)\cos^2(qz))}}.$$

(6)

In the presence of splay deformations, i.e. $\theta_a \neq \theta_b$, $\kappa_{\text{TSB}}(z)$ is a non-trivial periodic function of $z$. In Fig. 6a–b, we show the curvature $\kappa_{\text{SB}}(z)$ of the integral curves of the nematic director field of various splay-bend (SB) phases with $\theta_a = \theta_0$ and $\theta_b = 0$ from theory and simulations, along with their density profiles $\rho(z)$, measuring the probability of finding a particle at $z$. The periodicity of $\kappa_{\text{SB}}(z)$ agrees well with that of $\log \rho(z)$, i.e. minus the effective mean-field potential felt by the particles. If the

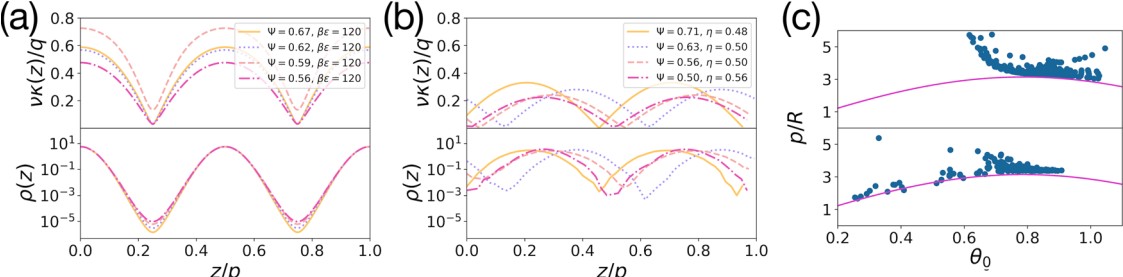

**Fig. 6 Rationalization of the spatial modulations of the density and the nematic director fields. a–b** Modulations of the curvature $\kappa(z)$ of the integral curves of the nematic director field (top), and of the logarithm of the density profile $\rho(z)$ (bottom) corresponding to minus the effective potential acting on the particles, in splay-bend smectic states (Sm$_{SB}$) predicted by our theory (**a**) and found in Monte Carlo simulations (**b**) for a system of hard curved spherocylinders of various curvatures $\Psi$ and inverse temperatures $\beta\epsilon$ or packing fractions $\eta$ as labelled. **c** Pitch length $p$ versus the conical angle $\theta_0$ of various twist-bend nematic (N$_{TB}$) phases of hard curved spherocylinders of various curvatures $\Psi \in [0.5, 2]$ from theory (top) and simulations (bottom), along with the relationship of Eq. (7) (pink line). Source data are provided as a Source Data file.

curvature of the integral curves of the nematic director field matches that of the particles, the particles can optimally align their shape to the local nematic director field, resulting into a lower potential energy or higher free volume and thus a higher local density $\rho(z)$, whereas geometric frustration arises when the curvature of the integral curve deviates from that of the particles, leading to a higher potential energy or lower free volume and hence a lower local density. We thus conclude that the presence of periodic density modulations in TSB and SB phases, and hence Sm$_{TSB}$ and Sm$_{SB}$ phases, can be explained geometrically by the coupling of particle curvature to the non-uniform bend deformations of the director field, but also by the coupling of density to splay deformations[27,28].

On the other hand, in the case of an N$_{TB}$ phase the translational symmetry of the elastic deformations is preserved, see Supplementary Note 2, and we expect the curvature of the integral curves of the nematic director field to be uniform. Intriguingly, for N$_{TB}$ phases with $\theta_a = \theta_b = \theta_0$ the integration of Eq. (5) can be carried out explicitly, yielding the following expression $\mathbf{r}_{TB}(z) = \mathbf{r}_{TB}(z_0) + \frac{\nu}{q}\sin\theta_0[\sin(qz) - \sin(qz_0)]\,\mathbf{i} - \frac{\nu}{q}\sin\theta_0[\cos(qz) - \cos(qz_0)]\,\mathbf{j} + \nu\cos\theta_0[z - z_0]\,\mathbf{k}$ for the integral curve of the nematic director field, i.e. a helix of period $p\nu\cos\theta_0$. Hence, we can impose that the integral curve has the same pitch $p$ as the N$_{TB}$ phase — or, equivalently, that $\mathbf{r}(z) \cdot \hat{z} = z$ — by setting $\nu = 1/\cos\theta_0$. As expected, the curvature of Eq. (6) reduces to a uniform value $\kappa_{TB}(z) = (q/\nu)\sin(\theta_0) = q\sin(2\theta_0)/2$ independent of $z$. Using the conjecture that particles tend to match their curvature with the one of the integral curves of the nematic director field, we impose $\kappa_{TB} = 1/R$, and obtain the simple relation

$$p = \pi R \sin(2\theta_0) \tag{7}$$

between the pitch $p$ and conical angle $\theta_0$ of an N$_{TB}$ phase and the particle curvature $R$. In Fig. 6c we test this simple relation against N$_{TB}$ phases from theory and simulations, finding that it describes the data remarkably well without any fit parameter. The most significant deviation is found for the theoretically predicted N$_{TB}$ states close to the I-N phase transition, where the pitch increases significantly across a small temperature range[24], a behavior not captured by Eq. (7) and our simulations. We remark here that the absence of an increase in the pitch in simulations may be caused by the finite size of the simulation box and the periodic boundary conditions. However, larger simulation boxes are beyond the limits of our computational resources. Figure 6c shows that the conical angle and the pitch vary in the range $\theta_0 \in [0.2, 0.9]$, and $p/R \in [1, 5]$. We thus find that the pitch length $p$ is on the order of the radius of curvature $R$ of a particle

independent of $L$, which can be used as a simple design rule and is consistent with the small pitch lengths observed for thermotropic bent-core mesogens[1–12]. We remark here that this phase is also called a polar twisted nematic instead of an N$_{TB}$ phase only because of its small pitch length[29,30]. In this work we prefer to classify the liquid crystal phases according to their symmetries. We also note that the smectic layer spacing $\lambda = p/2$ varies as $\lambda/L \in [0.32, 4.16]$. Hence, the unidentified Sm$_X$ phase with a smectic layer spacing of $\lambda \simeq 0.5L$ may perhaps be a Sm$_{TSB}$ or Sm$_{SB}$ phase[21].

**Discussion**

We introduce a generic nematic N$_{TSB}$ phase with twist, splay, and bend modulations in the director field which reduces to N, N$^*$, N$_{TB}$, and N$_{SB}$ phases in limiting cases. We use the nematic director field of this N$_{TSB}$ phase as a variational ansatz to develop a simple but comprehensive variational Maier–Saupe theory of periodically deformed nematic and smectic phases. We exploit this mean-field theory to predict the phase behavior of curved rods as a function of thermodynamic state and microscopic details, and find excellent agreement with simulations on hard curved spherocylinders. To characterise the symmetry and local structure of these modulated phases, we measure the scalar order parameter $S(z)$, nematic director field $\hat{\mathbf{n}}(z)$, polarisation vector $\hat{\mathbf{m}}(z)$, and density field $\rho(z)$ in simulations. We observe that the polarization is always anti-parallel to the director of the bend deformations, demonstrating that polarization and bend deformation are coupled by the flexo-electric effect originally proposed by Meyer[13]. We conclude that the mechanism behind these spatially modulated phases is the spontaneous polar ordering and bend deformations arising from the packing efficiency of curved particles. Our key findings are (1) that the N$_{TB}$ transforms into a Sm$_{SB}$ phase (or vice versa) via a gradual squeezing (or opening) of the precession cone, passing through an intermediate Sm$_{TSB}$ phase, and (2) that the elusive N$_{SB}$ phase is unstable with respect to the density-modulated Sm$_{SB}$ phase. Accurate free-energy calculations in computer simulations are required to determine the thermodynamic stability of the Sm$_{TSB}$ phase and the nature of the N$_{TB}$, Sm$_{TSB}$, and Sm$_{SB}$ phase transitions in the bulk phase diagram of hard curved spherocylinders.

Moreover, the agreement between the phase diagram determined by our simple mean-field theory for a thermotropic system and the one obtained from simulations for a lyotropic system, not only demonstrates the predictive power of our simple variational Maier–Saupe theory, but also provides strong evidence that the particle curvature is the driving force behind the topology of the spatial director-field modulations in the N$_{TB}$, Sm$_{TSB}$, and Sm$_{SB}$ phases. To rationalize this finding, we calculate the integral curves

of the nematic director field. We show that the curvature of these integral curves is periodic in the case of $Sm_{TSB}$ and $Sm_{SB}$ phases, and that the coupling of particle curvature to the non-uniform curvature of the director field leads to periodic density modulations. In the case of $N_{TB}$ phases, we derive an explicit expression for the uniform curvature of the nematic director field integral curves. By matching this curvature with that of the particles, we derive a simple relationship between the pitch and conical angle of the $N_{TB}$ phase and the microscopic particle curvature. We verify this simple relationship using theory and simulations.

In conclusion, our variational ansatz for a twist-splay-bend phase is a powerful tool for predicting, understanding and rationalising spatially modulated liquid crystal phases. Exploiting the generality of this variational ansatz in a generalized Maier–Saupe theory enabled us to predict not only the stability of twist-bend and splay-bend phases, but also the orientational order parameters, pitch and conical angle as a function of the thermodynamic state and microscopic details of the particles, see Supplementary Note 5. This variational ansatz can also be exploited in Landau–de Gennes and Oseen–Frank theories of spatially modulated phases. Further improvements of the Maier–Saupe theory such as introducing biaxiality[31], extending the description from prolate to oblate liquid crystals, or generalizing the variational ansatz for spatial modulations from 1D to 2D and 3D to describe polar blue phases[32], may lead to a generic theoretical framework of modulated liquid crystal phases.

## Methods

*Variational Maier–Saupe theory.* To solve our generalized Maier–Saupe theory, we minimize the free energy in Eq. (4) with respect to the variational parameters of the director- and density-field. Calculating the free-energy difference per particle $\beta\Delta F/N$ is trivial except for the partition function $Q$, i.e. an integral of the form

$$I = \int \mathrm{d}\Omega \int_0^p \mathrm{d}Z\, f(Z,\Omega)$$
$$= \int_0^{2\pi} \mathrm{d}\alpha \int_0^{\pi} \sin\beta \mathrm{d}\beta \int_0^{2\pi} \mathrm{d}\gamma \int_0^p \mathrm{d}Z\, f(Z,\Omega), \quad (8)$$

with $f(Z,\Omega) = \exp[-\beta U(Z,\Omega)]$. We assume that the function $f(Z,\Omega)$ is a periodic function in $Z$ with period $p$. To evaluate the integrals of the form (8), we first transform the original space of integration to a 4-dimensional hypercube $[-1,1]^4$ using the transformation $\alpha = \pi(\xi+1), \cos\beta = \eta, \gamma = \pi(\mu+1), Z = \frac{p}{2}(\psi+1)$ with Jacobian $|J| = p\pi^2/2$. The integral in Eq. (8) becomes

$$I = \int_{-1}^1 \mathrm{d}\xi \int_{-1}^1 \mathrm{d}\eta \int_{-1}^1 \mathrm{d}\mu \int_{-1}^1 \mathrm{d}\psi\, \frac{p\pi^2}{2} f(Z(\psi),\Omega(\xi,\eta,\mu)), \quad (9)$$

which we solve using an $\mathcal{N}$-points Gauss-Legendre quadrature. If $\{\xi_i\}, \{\eta_i\}, \{\mu_i\}$, and $\{\psi_i\}$ are the $\mathcal{N}$ node points in $[-1,1]$, i.e. the $\mathcal{N}$ roots of the $\mathcal{N}$-th Legendre polynomial, and $\{w_{\xi,i}\}, \{w_{\eta,i}\}, \{w_{\mu,i}\}$, and $\{w_{\psi,i}\}$ are the associated weights, we can approximate the integral $I$ of Eq. (9) by

$$I \approx \sum_{i=1}^{\mathcal{N}} \sum_{j=1}^{\mathcal{N}} \sum_{k=1}^{\mathcal{N}} \sum_{l=1}^{\mathcal{N}} w_{\xi,i} w_{\eta,j} w_{\mu,k} w_{\psi,l} \frac{p\pi^2}{2} f(Z(\psi_l),\Omega(\xi_i,\eta_j,\mu_k)). \quad (10)$$

Using the approximation of Eq. (10) and $f(Z(\psi), \Omega(\xi,\eta,\mu)) = \exp[-\beta U(Z(\psi),\Omega(\xi,\eta,\mu))]$, we evaluate the partition function in Eq. (4) using an $\mathcal{N} = 32$ points Gauss-Legendre integration.

Once $Q$ is calculated, we minimize $\beta\Delta F/N$ at given $\beta\epsilon$ in the 6-dimensional space of parameters $S$, $\tau$, $n$, $\theta_a$, $\theta_b$ and $q = 2\pi/p$ by

means of a Covariance Matrix Adaptation Evolution Strategy (CMA-ES) using the Python library in https://pypi.org/project/cma/.

*Monte Carlo simulations.* We study the phase behavior of a system consisting of hard curved spherocylinders using Monte Carlo simulations in the $NPT$ ensemble, i.e. at fixed number of particles $N$, pressure $P$ and temperature $T$. We employ an orthorhombic simulation box of sides $L_x$, $L_y$, and $L_z$ and apply periodic boundary conditions. We perform a sequence of MC cycles consisting of $N+1$ MC moves. Each MC move consists of a particle move with probability $\sim N/(N+1)$, and a volume move with probability $\sim 1/(N+1)$. In a particle move, a random roto-translation of a randomly picked particle is proposed and accepted if it does not generate overlaps with other particles. In a volume move, a random variation of a random side of the simulation box is proposed, and the system is compressed or expanded accordingly. If the compression/expansion does not generate overlaps between the particles, the move is accepted with a probability

$$\mathrm{acc}(V \to V') = \min\left(1, \left(\frac{V'}{V}\right)^N e^{-\beta P\Delta V}\right), \quad (11)$$

where $\Delta V = V' - V$ denotes the change in volume. We perform simulations on a system of $N = 2048$ hard curved spherocylinders, and initialize all simulations from a nematic configuration. We measure a wide set of observables during the simulation, like density, uniaxial nematic order parameters, smectic order parameters, etc. When the system reaches equilibrium as monitored from the observables, we characterize the system's configuration. We find that $10^8$ MC cycles are typically sufficient for equilibration.

To study a system of hard curved spherocylinders in a gravitational field, we perform Monte Carlo simulations in an $NVT$ ensemble, i.e. we fix the number of particles $N = 8192$, volume $V$, and temperature $T$. We implement a hard wall at the bottom at $z = 0$ and apply periodic boundary conditions in the $x$− and $y$− direction. Each MC cycle consists of $N$ attempts to randomly rotate and translate a randomly selected particle. If a particle roto-translation leads to an overlap with one of the particles or with the hard wall, the move is rejected, otherwise it is accepted with a probability

$$\mathrm{acc}(z \to z') = \min(1, e^{-\beta\Delta U_g}), \quad (12)$$

with $\Delta U_g = (z' - z)/l_g$ the change in potential energy due to gravity, $z$ and $z'$ the old and new $z$-coordinate of the displaced particle, and $l_g$ the gravitational length of the system.

## Data availability
The source data from Figs. 3, 5b, 6, Supplementary Figs. 1, 2, 3, 7, 8, 9, 11, 12, 13, 14, and 15 are provided in the source data file. All the other relevant data associated with this research is available upon request. Source data are provided with this paper.

## Code availability
The simulation and analysis codes associated with this research are available upon request.

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

## Acknowledgements

The authors acknowledge financial support from the EU H2020-MSCA-ITN- 2015 project 'MULTIMAT' (Marie Sklodowska-Curie Innovative Training Networks) [project number: 676045].

## Author contributions

M.D. initiated this work on the phase behavior of curved rods and supervised M.C. M.C. generalized the Maier–Saupe theory to curved rods and performed the theoretical calculations and the computer simulations on hard curved spherocylinders. M.C. and M.D. analysed and discussed the results, and co-wrote the paper.

## Competing interests

The authors declare no competing interests.
