## [Peer Review File · Nature Communications]

REVIEWER COMMENTS

Reviewer #1 (Remarks to the Author):

This is a very nice paper, well-written and well-presented and presenting both theoretical and simulation evidence in support of the existence of a new liquid crystalline phase. Many high profile papers have been written about twist-bend and splay-bend nematic phases so this general area might be considered to be a hot topic. This paper argues that there is something like a "missing-link" between the structures of these two known phases and which, unlike normal nematic phases, has a density modulation like a smectic. I thus believe that this result is of sufficient general interest and novelty to be publishable in Nature Communications and that it will inspire new research in this area.

The fact that this new phase is found both by simulation and by the solution of a Maier-Saupe equation adds much weight to the conclusion. So far as I can tell, the theoretical approach is sound and the analysis looks to be well done. I have a few queries which I would like the authors to address, but they are of a relatively minor nature.

1) While the authors have tried very hard to illustrate what the phases in question look like, I confess I still struggle to have an intuitive picture of the TSB phase. The pictures with curved spherocylinders look lovely but are too dense to show what is happening while fig 2, again for all the effort that went into it, is still hard to grasp. This may be impossibly difficult, but is there a way of making the curved spherocylinder picture more schematic, just so the reader can get a clear idea of what is twisting around what?

2) I am still confused by the expression for the director field given in this paper and by Dozov (see, e.g., under fig 3). I understand that numerically they are very similar, but that comes over as luck! Is one right and the other wrong or is it more the case that there is some freedom for arbitrariness here?

3) This again might be impossible, but could the authors explain why there have to be density fluctuations in the TSB phase? Is this because the structure can in principle exist at a uniform density but would prefer not to, or that it is geometrically impossible for such a structure to have a uniform density? One reason for asking is that the related structure, found by applying a field to a TB phase (e.g. ref. 20), was not predicted to be associated with such a fluctuation.

4) The article needs a few details about the simulation methodology. While much of this is appropriate for Supplementary Information, the reader should have an idea of box size, number of particles, the initial configurations from which the system evolved and the number of cycles for equilibration and production.

5) The authors should comment in the main paper on the re-entrant SB phase, shown in Fig 2 in the Maier-Saupe plot at high bend angle, which is not observed in the simulation. That seems to be the main qualitative difference between theory and simulation.

6) Bent core molecules are known to form curious, frustrated smectic like structures (banana phase, dark conglomerate, etc.). Because of the length scales involved, however, a simulation of such a phase may simply show something looking like a normal smectic. Could the authors briefly comment on where such phases might lie with respect to the phase diagram they plot and also on how sure they are that their simulations might not miss frustration effects because of the simulation box size?

I think these comments are easy to address (except for the challenging one of how best to sketch a TSB phase!), but I think the response would help the reader and strengthen the paper.

A. J. Masters

Reviewer #2 (Remarks to the Author):

Dear Editors,

In 2010, it was demonstrated experimentally that a series of symmetric achiral hydrocarbon linked mesogenic dimers exhibit, apart from the conventional uniaxial nematic phase, a second, distinct, nematic state with broken mirror symmetry. Since then, there has been intense research concerning the physics of this novel mode of molecular self organization. The topic is of high interest not only in connection with the physics of liquid crystals but also with the physics of self-organization in soft matter in general.

Despite the intense scientific efforts there is not a unified and broadly accepted interpretation on the microscopic origins of a) the very existence of a nematic-nematic phase transition in single component bent-core systems, b) the broken chiral symmetry in a positionally disordered liquid phase composed by achiral molecules and c) the structure and symmetries on the nanometer scale of the new nematic phase.

Here, the authors using MC computer simulations study the phase behavior of sterically interacting curved rods (spherocylinders) of C_{2v} symmetry. They demonstrate that such systems, depending on the curvature of the particles, may exhibit two nematic phases. The low-density phase is a common uniaxial nematic while in the high-density phase the direction of the local (nematic like) molecular ordering twists continuously about a macroscopic axis resembling the structure predicted some decades ago by R.B. Meyer, known as NTB (Twist-Bend-Nematic). There are also previous simulation works either with rigid or with flexible particles demonstrating the same phase sequence which should be cited. At even higher densities the systems exhibit several smectic-like phases that resemble proposed structures of nematics with spontaneous splay-bend (NSB) or with spontaneous twist-splay-bend (NTSB) deformations of the director.

The authors attempt to provide a link between the inherent molecular curvature and the local and macroscopic(?) structure of the possible modulated nematic (or smectic) phases in such systems. To do this they use a simple mean-field-like approximation to describe the molecular ordering (purely orientational or mixed positional-orientational) of the curved particles under the influence of an ad hoc director field which is an extension of related proposals by Meyer (1973) and Dozov (2001). It should be mentioned, however, that the director deformations proposed by Meyer and Dozov are based on a continuum theory and therefore on the assumption that the distances over which significant variations of the nematic field occur are much larger than the molecular dimensions. Clearly this is not the case in the TB, SB and TSB phases reported in the submitted work.

The paper concludes that the NTB phase is thermodynamically stable and that SB and TSB deformations are associated inevitably with (topology-driven) density modulations and therefore the corresponding phases cannot be nematics. Given the well-known fact that states of pure constant splay or bend are not possible in a continuous three-dimensional object, it is not surprising that phases with such deformations will be accompanied with either domain formation and/or density modulations.

Although the results of the simulations are very interesting, their interpretation in terms of deformations of a nematic director field needs more convincing justification. The authors should analyze and discuss in depth the following issues:

- 1) Both Meyer and Dozov assume explicitly that the (twist-bend or splay bend) deformations of the nematic field is at length scales much larger than the molecular dimensions. For instance, the helical pitch in the NTB phase is predicted to be of the order of hundreds molecular lengths. In this work the simulation results indicate clearly that the pitch is of the order of a single molecular length ($p/R=1...5$). Thus, the calculated pitch is in the molecular length scale and certainly does not correspond to a macroscopic modulation, as mentioned in the text several times.
- 2) Spontaneous polar ordering is assumed to be the very origin of the TB deformation of the NTB phase as proposed by Meyer. In the submitted manuscript this type of molecular ordering is not examined. This is somehow unexpected since visual inspection of the presented snapshots suggests that the steric molecular dipoles (defining the C_2 molecular axis) within thin slabs perpendicular to Z-axis in the NTB, SB and TSB states exhibit, at least to some extent, polar ordering. This kind of order, the corresponding polar director and its correlations along the Z-axis, if present, should be quantified and reported.

In conclusion, in its present form the paper does not clarify or shed new light on the fundamental questions regarding the low temperature nematic phase exhibited by curved mesogens. Despite the wealth of original simulation results, an important question remains open: Is the symmetry and local structure of the second nematic phase of the curved rods in accordance with the conjectures of Meyer, of Dozov, or to none of them? In the latter case does it correspond to a distinct nematic state with the polar ordering of the molecules being the driving force for the stabilization of the (molecular scale) modulations?

The reported NTB nature of the high packing fraction nematic state of the hard curved cylinders is an a priori assumption and not a conclusion based on solid arguments supported by calculated quantities like polar ordering and corresponding correlations. Finally, the mix up between the microscopic and the (assumed) macroscopic length scales, renders the whole discussion on the lack of thermodynamically stable NSB and NTSB phases in the simulations, rather weakly connected to the (correct) arguments that topology drives the density modulations is SB and TSB phases. For these reasons I cannot recommend publication of the paper in its present form.

Reviewer #3 (Remarks to the Author):

These are exciting results, more exciting than the authors admit: they have demonstrated new smectic phases which they call TSB and SB. The geometric argument that splay-bend and twist-splay-bend requires density modulation is important. This must arise through the coupling of splay to density in polymer nematics, for instance (see V.G. Taratuta, R.B. Meyer. *Liq. Cryst.*, 2 (1987), p. 373). Perhaps this connection could be made since it is important from a symmetry point of view.

However, I do not understand the language in the section on "Rationalising modulated liquid crystal phases in terms of pseudomanifolds". The construction in equation (5) gives the integral curves of the director field, not two-dimensional surfaces. However, the following step where the curvature of the integral curves is shown to be periodic is the essential point: in the N_TB phase there is no curvature modulation to couple with the molecular shape -- only the Sm_TSB phase has a periodic curvature. The pseudomanifolds are constructed via the curvature modulation, not from the integral curves.

Minor points:

- 1) Why not name the phases $Sm_{\{TSB\}}$ and $Sm_{\{SB\}}$ since they are smectics?
- 2) The phrase "chiral symmetry breaking" is commonly used but is also wrong: chirality is an absence of symmetry so it would be more appropriate to call it "achiral symmetry breaking".

This is an important result in the understanding of these new phases that connects molecular shape to phase behavior. The authors should revise the language and discussion but this article should be published after that happens.

Reviewer #1

This is a very nice paper, well-written and well-presented and presenting both theoretical and simulation evidence in support of the existence of a new liquid crystalline phase. Many high profile papers have been written about twist-bend and splay-bend nematic phases so this general area might be considered to be a hot topic. This paper argues that there is something like a "missing-link" between the structures of these two known phases and which, unlike normal nematic phases, has a density modulation like a smectic. I thus believe that this result is of sufficient general interest and novelty to be publishable in Nature Communications and that it will inspire new research in this area.

The fact that this new phase is found both by simulation and by the solution of a Maier-Saupe equation adds much weight to the conclusion. So far as I can tell, the theoretical approach is sound and the analysis looks to be well done. I have a few queries which I would like the authors to address, but they are of a relatively minor nature.

We thank the referee for her/his words of appreciation, and for the useful comments that helped us to improve the clarity of the manuscript's scientific message.

1) While the authors have tried very hard to illustrate what the phases in question look like, I confess I still struggle to have an intuitive picture of the TSB phase. The pictures with curved spherocylinders look lovely but are too dense to show what is happening while fig 2, again for all the effort that went into it, is still hard to grasp. This may be impossibly difficult, but is there a way of making the curved spherocylinder picture more schematic, just so the reader can get a clear idea of what is twisting around what?

We agree with the Referee that it is not easy to get a clear picture of the twist-splay-bend nematic phase by looking at the 2D schematics in Figure 2, and the computer simulation snapshots in Figure 4. To illustrate better the TSB phase, we added in Figure 1 a schematic of the arrangement of curved rods in the TSB phase, as well as top views of these schematics for the N_{TB} , N_{TSB} , and N_{SB} phases, yielding a clearer 3D picture of the continuous transformation of the N_{TB} phase to the N_{SB} phase.

In the revised Supplemental Materials we reproduce the simulation configurations of Figure 4, but also include the top view (xy-plane) of the simulation configuration and a schematic of the precession cone in order to visualise better the 3D arrangement of the curved rods.

2) I am still confused by the expression for the director field given in this paper and by Dozov (see, e.g., under fig 3). I understand that numerically they are very similar, but that comes over as luck! Is one right and the other wrong or is it more the case that there is some freedom for arbitrariness here?

The Referee raises an interesting and valid point concerning the different expressions for the nematic director field of a splay-bend nematic phase reported in this paper and by Dozov. Both expressions are variational ansatzes for the nematic director field of which the cone angle and pitch length are the variational parameters. Minimizing the free energy with respect to these variational parameters gives the equilibrium phase. However, we cannot exclude that a full minimization of the free energy without assuming a variational ansatz for the nematic director field might give a slightly different nematic director field with an even lower free energy. Hence, we agree with the referee that there is some freedom for arbitrariness here.

Having said that, the nematic director field of the SB phases measured in our simulations are well-fitted by both expressions, see Figure S15. In addition, the free energy obtained with our expression in our Maier-Saupe theory gives lower free-energies than Dozov's expression for $\theta_0 > 0.9$, where the difference between the two expressions becomes significant (see Figure S1).

We state this more clearly in the revised Supplemental Materials.

Finally, we make the following remark. In his original work, Dozov assumed that the bend elastic constant K_3 might become negative in systems of banana-shaped particles. Within this assumption, the usual second-order expansion of the elastic free energy is not bounded from below, and a large number of fourth-order elastic terms with unknown elastic constants need to be included to obtain an equilibrium state with finite bend. Dozov limited his analysis to one-dimensional deformations (*i.e.* $\mathbf{n}(\mathbf{r}) = \mathbf{n}(z)$), and considered only terms compatible with uniaxial nematic symmetry, and derived the following fourth-order expression for the elastic free energy of deformations

$$F = \frac{1}{2} \{ K_1[\mathbf{n} \cdot (\nabla \cdot \mathbf{n})]^2 + K_2[\mathbf{n} \times (\nabla \times \mathbf{n}) + K_3[\mathbf{n} \cdot (\nabla \times \mathbf{n})]^2]^2 \} + \frac{1}{4} \{ C_1[(n_l n_k)']^2 + 2C_2[(n_3 n_k)']^2 + C_3[(n_3)']^2 \},$$

where $'$ indicates the second derivative with respect to z and the summation runs over the repeated subscripts.

Assuming a negative K_3 , a ground state of finite bend must exist. Since a pure bend deformation cannot uniformly fill the 3D space, Dozov postulated the existence of states with a finite bend deformation and complementary deformations of twist or splay, *i.e.* the twist-bend and splay-bend nematic phases. For the N_{SB} phase, Dozov *supposes* the expression $\mathbf{n}(z) = (\sin(\theta_0 \sin(qz)), 0, \cos(\theta_0 \sin(qz)))$, and derives the values of θ_0 and q by minimising the free energy within the assumption $\theta_0 \ll 1$, thereby neglecting terms of higher order in θ_0 .

As Dozov's and our expression for the nematic director field $\mathbf{n}(z)$ are indistinguishable for $\theta_0 \ll 1$, *i.e.* within Dozov's theoretical assumptions, we expect our expression to be also a solution of Dozov's theoretical model.

3) *This again might be impossible, but could the authors explain why there have to be density fluctuations in the TSB phase? Is this because the structure can in principle exist at a uniform density but would prefer not to, or that is is geometrically impossible for such a structure to have a uniform density? One reason for asking is that the related structure, found by applying a field to a TB phase (e.g. ref. 20), was not predicted to be associated with such a fluctuation.*

We thank the Referee for raising this interesting and relevant question. To answer this question, we refer to Referee #3, who pointed us to an early article by Meyer showing that splay deformations are necessarily associated with density modulations. Both the TSB and TB phase display non-uniform splay and bend deformations in the nematic director field. The appearance of density modulations can be explained geometrically by the coupling of the particle shape to the non-uniform curvature of the nematic director field, but also by the coupling of density to splay. We clarify this in more detail in the revised manuscript.

In the abstract:

We find that N_{TB} phases transition towards splay-bend (SB) phases via intermediate twist-splay-bend (TSB) phases exhibiting periodic density modulations which we explain by the coupling of particle curvature to the non-uniform curvature of the nematic director field.

In the main text:

This confirms that splay deformations of the nematic director field are inherently associated with the

appearance of density modulations, as already speculated by De Gennes and Meyer four decades ago^{25,26}.

We thus conclude that the presence of periodic density modulations in TSB and SB phases can be explained geometrically by the coupling of particle curvature to the non-uniform bend deformations of the director field, but also by the coupling of density to splay deformations^{25,26}.

We show that the curvature of these integral curves is periodic in the case of TSB and SB phases, and that the coupling of particle curvature to the non-uniform curvature of the director field leads to periodic density modulations.

We also note that the theory in Ref. [20] is a macroscopic phenomenological Landau-De Gennes theory which does not account for density modulations. In order to account for density modulations, one should resort to a Landau-de Gennes theory for lyotropic systems using a Q-tensor expansion of the chemical-potential dependent grand potential, see e.g. C. Anzivino et al., J. Chem. Phys. 152, 224502 (2020).

4) The article needs a few details about the simulation methodology. While much of this is appropriate for Supplementary Information, the reader should have an idea of box size, number of particles, the initial configurations from which the system evolved and the number of cycles for equilibration and production.

We refer the Referee to the Methods section of the manuscript, where all the details on the Monte Carlo simulations in bulk and in sedimentation were already reported.

5) The authors should comment in the main paper on the re-entrant SB phase, shown in Fig 2 in the Maier-Saupe plot at high bend angle, which is not observed in the simulation. That seem to be the main qualitative difference between theory and simulation.

We already comment in the current version of the main paper on the re-entrant TSB phase:

“Moreover, we find two distinct regions of splay-bend phases, one at low particle curvature and one at high curvature, the latter transforming into a re-entrant twist-splay-bend phase with increasing $\beta\epsilon$.”

However, following the Referee’s suggestion, in the updated paper we also stress that this re-entrant TSB phase is not observed in the simulated phase diagram.

We note that the main qualitative difference is the absence of the re-entrant Sm_{TSB} phase in the simulated phase diagram.

Finally, we mention that due to hysteresis and slow equilibration, it is impossible to accurately determine the regions of stability and the first- or second-order nature of the phase transitions of the N_{TB} , Sm_{TSB} , and Sm_{SB} phases in simulations of hard curved spherocylinders. To show this more clearly, we modified the colouring and partitioning of the phase diagram of Figure 3b and state this more carefully in the revised manuscript.

In the main text:

It is important to mention here that due to hysteresis effects and slow equilibration it is impossible to accurately determine the regions of stability and the first- or second-order nature of the phase transitions of the N_{TB} , Sm_{TSB} , and Sm_{SB} phases in simulations of hard curved spherocylinders. However, the simulations consistently show that the phase transformation from a N_{TB} to a Sm_{SB} phase proceeds via an intermediate Sm_{TSB} phase as twist deformations are gradually replaced by splay deformations of the nematic director field. For example, in Figure 4 we show typical configurations of a Sm_{SB} phase, a

Sm_{TSB} phase, and a N_{TB} phase along an expansion of a system of hard curved spherocylinders of length $L/D = 19$ and opening angle $\Psi = 1.31$ from packing fraction $\eta = 0.406$ to packing fraction $\eta = 0.367$.

In the conclusions:

Accurate free-energy calculations in computer simulations are required to determine the thermodynamic stability of the Sm_{TSB} phase and the nature of the N_{TB} , Sm_{TSB} , and Sm_{SB} phase transitions in the bulk phase diagram of hard curved spherocylinders.

6) Bent core molecules are known to form curious, frustrated smectic like structures (banana phase, dark conglomerate, etc.). Because of the length scales involved, however, a simulation of such a phase may simply show something looking like a normal smectic. Could the authors briefly comment on where such phases might lie with respect to the phase diagram they plot and also on how sure they are that their simulations might not miss frustration effects because of the simulation box size?

The Referee raises an interesting question concerning the observation of various non-equilibrium phases observed for bent-core molecules. The focus of our work is, however, on the equilibrium phase behavior of banana-shaped particles, and the good agreement between the theory and simulations provides confidence that our results on the phase behavior are reliable, even though we only allowed for 1-dimensional modulations in the nematic director field in the Maier-Saupe theory, thereby ignoring the possibility of higher-dimensional modulations as observed in the dark conglomerate and polar blue phases.

In our view, the dark conglomerate phase and polar blue phases of banana-shaped particles are inherently non-equilibrium phases, arising from the formation of line and point defects in the nematic texture. In order to avoid the formation of defects and facilitate the formation of equilibrium spatially modulated nematic phases, we always initialize our simulations in a uniaxial nematic phase, see Methods sections. When we rapidly compress an isotropic phase, multiple nematic nuclei are formed with different orientations throughout the system, leading to the formation of line and point defects and resulting into polar blue phases. Below, we exemplarily show a typical configuration of a polar blue phase which is metastable with respect to the equilibrium twist-bend nematic phase.

Reviewer #2

Dear Editors,

In 2010, it was demonstrated experimentally that a series of symmetric achiral hydrocarbon linked mesogenic dimers exhibit, apart from the conventional uniaxial nematic phase, a second, distinct, nematic state with broken mirror symmetry. Since then, there has been intense research concerning the physics of this novel mode of molecular self organization. The topic is of high interest not only in connection with the physics of liquid crystals but also with the physics of self-organization in soft matter in general.

Despite the intense scientific efforts there is not a unified and broadly accepted interpretation on the microscopic origins of a) the very existence of a nematic-nematic phase transition in single component bent-core systems, b) the broken chiral symmetry in a positionally disordered liquid phase composed by achiral molecules and c) the structure and symmetries on the nanometer scale of the new nematic phase.

Here, the authors using MC computer simulations study the phase behavior of sterically interacting curved rods (spherocylinders) of C_{2v} symmetry. They demonstrate that such systems, depending on the curvature of the particles, may exhibit two nematic phases. The low-density phase is a common uniaxial nematic while in the high-density phase the direction of the local (nematic like) molecular ordering twists continuously about a macroscopic axis resembling the structure predicted some decades ago by R.B. Meyer, known as NTB (Twist-Bend-Nematic). There are also previous simulation works either with rigid or with flexible particles demonstrating the same phase sequence which should be cited. At even higher densities the systems exhibit several smectic-like phases that resemble proposed structures of nematics with spontaneous splay-bend (NSB) or with spontaneous twist-splay-bend (NTSB) deformations of the director.

We thank the Referee for a careful reading of our paper and for the insightful comments and helpful suggestions. We agree with the referee that we should have cited the previous simulation works demonstrating the same phase sequence. We have added the references in the revised manuscript.

On the other hand, computer simulations demonstrate the existence of $N_{\{TB\}}$ phases for both rigid and flexible banana-shaped molecules [Memmer 2002, Chen 2013, Greco 2015, Chiappini 2019].

The authors attempt to provide a link between the inherent molecular curvature and the local and macroscopic(?) structure of the possible modulated nematic (or smectic) phases in such systems. To do this they use a simple mean-field-like approximation to describe the molecular ordering (purely orientational or mixed positional-orientational) of the curved particles under the influence of an ad hoc director field which is an extension of related proposals by Meyer (1973) and Dozov (2001). It should be mentioned, however, that the director deformations proposed by Meyer and Dozov are based on a continuum theory and therefore on the assumption that the distances over which significant variations of the nematic field occur are much larger than the molecular dimensions. Clearly this is not the case in the TB, SB and TSB phases reported in the submitted work.

We agree with the referee that the expressions proposed by Meyer and Dozov are based on a continuum theory, which may break down when the variations of the nematic director field are strong as is the case for the TB, SB, and TSB phases reported in our work. However, we show that the nematic

director field of these phases measured in our simulations are well-fitted by these expressions, see e.g. Figure 5 and Figure S15, showing that the continuum description is valid on this small scale.

We also note that this argument was put forward very recently by Samulski, Vanakaras, and Photinos in a paper entitled “*The twist-bend nematic: A case of mistaken identity*” in *Liq. Cryst.* (2020), which ignited a heated debate, see e.g. the paper entitled “*Setting things straight in ‘The twist-bend nematic: a case of mistaken identity’*” by Dozov and Luckhurst in the same issue of *Liq. Cryst.* (2020).

Without entering this debate, we merely show here that the director fields obtained from simulations are well-described by the expressions from continuum theory.

The paper concludes that the NTB phase is thermodynamically stable and that SB and TSB deformations are associated inevitably with (topology-driven) density modulations and therefore the corresponding phases cannot be nematics. Given the well-known fact that states of pure constant splay or bend are not possible in a continuous three-dimensional object, it is not surprising that phases with such deformations will be accompanied with either domain formation and/or density modulations.

Although the results of the simulations are very interesting, their interpretation in terms of deformations of a nematic director field needs more convincing justification. The authors should analyze and discuss in depth the following issues:

1) Both Meyer and Dozov assume explicitly that the (twist-bend or splay bend) deformations of the nematic field is at length scales much larger than the molecular dimensions. For instance, the helical pitch in the NTB phase is predicted to be of the order of hundreds molecular lengths. In this work the simulation results indicate clearly that the pitch is of the order of a single molecular length ($p/R=1...5$). Thus, the calculated pitch is in the molecular length scale and certainly does not correspond to a macroscopic modulation, as mentioned in the text several times.

We thank the Referee for her/his appreciation of our simulation results. We agree with the Referee that the spatial modulations of the TB, TSB, and SB phases in our work are on the order of a few particle lengths. In order to avoid confusion, we remove any reference to “microscopic” and “macroscopic”, to indicate “particle details” and the length scales associated to the modulations of the director field in the revised manuscript.

2) Spontaneous polar ordering is assumed to be the very origin of the TB deformation of the NTB phase as proposed by Meyer. In the submitted manuscript this type of molecular ordering is not examined. This is somehow unexpected since visual inspection of the presented snapshots suggests that the steric molecular dipoles (defining the C2 molecular axis) within thin slabs perpendicular to Z-axis in the NTB, SB and TSB states exhibit, at least to some extent, polar ordering. This kind of order, the corresponding polar director and its correlations along the Z-axis, if present, should be quantified and reported.

We agree with the Referee that a proper description of NTB, SB, and TSB phases should include not only the second-rank $\mathbf{Q}(\mathbf{r})$ tensor but also the polarisation vector $\mathbf{m}(\mathbf{r})$, which is perpendicular to the director $\mathbf{n}(\mathbf{r})$ and to the helical axis, but anti-parallel to the bend vector $\mathbf{b}(\mathbf{r})$. We follow the suggestion of the Referee, and plot the polarisation vector $\mathbf{m}(\mathbf{r})$ in the revised Supplemental Material of exemplary twist-bend, twist-splay-bend, and splay-bend states along with the bend vector $\mathbf{b}(\mathbf{r})$. We show that the polarization vector $\mathbf{m}(\mathbf{r})$ is always anti-parallel to the bend vector $\mathbf{b}(\mathbf{r})$, which is consistent with Meyer's picture that due to the bend flexoelectric effect spontaneous polar ordering is coupled to bend deformations in the nematic director field, resulting into a N_{TB} , SB, and TSB phases.

In the abstract:

[...], and demonstrate that the formation of these modulated phases is driven by bend deformations coupled to polar ordering spontaneously arising from packing constraints of banana-shaped particles.

In the main text:

To fully characterise the phases in Figure 4 we plot the scalar order parameter $S(z)$, the nematic director field $\mathbf{n}(z)$, and the density field $\rho(z)$ in the Supplemental Material. In addition, we also plot the polarisation vector $\mathbf{m}(z)$ along with the bend distortions in the director field described by the bend vector $\mathbf{b}(z) = \mathbf{n}(z) \times (\nabla \times \mathbf{n}(z)) / |\mathbf{n}(z) \times (\nabla \times \mathbf{n}(z))|$. We clearly observe that $\mathbf{m}(z)$ is always anti-parallel to $\mathbf{b}(z)$, demonstrating that these modulated phases are driven by bend deformations coupled to polar ordering spontaneously arising from the packing constraints of banana-shaped particles.

To characterise the symmetry and local structure of these modulated phases, we measure the scalar order parameter $S(z)$, nematic director field $\mathbf{n}(z)$, polarisation vector $\mathbf{m}(z)$, and density field $\rho(z)$ in simulations. We observe that the polarization is always anti-parallel to the director of the bend deformations, demonstrating that polarization and bend deformation are coupled by the flexo-electric effect originally proposed by Meyer¹³. We conclude that the mechanism behind these spatially modulated phases is the spontaneous polar ordering and bend deformations arising from the packing efficiency of curved particles.

We added a new section on polar order and bend deformations in the revised SM.

In conclusion, in its present form the paper does not clarify or shed new light on the fundamental questions regarding the low temperature nematic phase exhibited by curved mesogens. Despite the wealth of original simulation results, an important question remains open: Is the symmetry and local structure of the second nematic phase of the curved rods in accordance with the conjectures of Meyer, of Dozov, or to none of them? In the latter case does it correspond to a distinct nematic state with the polar ordering of the molecules being the driving force for the stabilization of the (molecular scale) modulations?

The reported NTB nature of the high packing fraction nematic state of the hard curved cylinders is an a priori assumption and not a conclusion based on solid arguments supported by calculated quantities like polar ordering and corresponding correlations. Finally, the mix up between the microscopic and the (assumed) macroscopic length scales, renders the whole discussion on the lack of thermodynamically stable NSB and NTSB phases in the simulations, rather weakly connected to the (correct) arguments that topology drives the density modulations is SB and TSB phases. For these reasons I cannot recommend publication of the paper in its present form.

In the revised manuscript, we have analysed in detail the symmetry and local structure of these spatially modulated phases which are characterised by a Q-tensor with periodic twist-bend and splay-bend distortions in the director field, as well as a polarization vector \mathbf{P} which is perpendicular to the local nematic director and parallel to the bend vector in agreement with the phases proposed by Meyer and Dozov. Within our understanding, Meyer's approach based on the flexoelectric coupling and Dozov's theory based on the negative elastic bend constant are two different but not mutually exclusive approaches to describe the same phenomenon, *i.e.* the stability of these spatially modulated phases. This is confirmed by the fact that Meyer's theory can be mapped onto Dozov's framework by introducing renormalised effective elastic constants that account for the flexoelectric effect and polarization [A. Jàkli et al., *Rev. Mod. Phys.*, **90** (2018)].

In conclusion, the symmetry and local structure of the twist-bend and splay-bend nematic phases reported in our work is consistent with the ones predicted by both Meyer and Dozov, as confirmed by the fact that the nematic director and polarization fields measured in simulations are well-fitted by the expressions reported by both Meyer and Dozov in their original works, see e.g. Figure S15. In addition, we show that the polarization and bend deformations are coupled by the flexoelectric effect proposed by Meyer, demonstrating that the physical mechanism behind these modulated phases is the spontaneous polar ordering and bend deformations arising from the packing efficiency of curved particles.

Reviewer #3

These are exciting results, more exciting than the authors admit: they have demonstrated new smectic phases which they call TSB and SB. The geometric argument that splay-bend and twist-splay-bend requires density modulation is important. This must arise through the coupling of splay to density in polymer nematics, for instance (see V.G. Taratuta, R.B. Meyer. Liq. Cryst., 2 (1987), p. 373). Perhaps this connection could be made since it is important from a symmetry point of view.

We thank the referee for appreciating our work and recommending publication in Nature Communications. We thank the referee for pointing us to this reference, which is an important argument for the required density modulations in splay-bend and twist-splay-bend phases. We have included the reference in the revised manuscript (see also the answer to Referee #1).

However, I do not understand the language in the section on "Rationalising modulated liquid crystal phases in terms of pseudomanifolds". The construction in equation (5) gives the integral curves of the director field, not two-dimensional surfaces. However, the following step where the curvature of the integral curves is shown to be periodic is the essential point: in the N_TB phase there is no curvature modulation to couple with the molecular shape -- only the Sm_TSB phase has a periodic curvature. The pseudomanifolds are constructed via the curvature modulation, not from the integral curves.

We agree with the Referee that the term "pseudomanifold" to denote the integral curves of the director field is confusing. We replaced the term "pseudomanifold" with the term "integral curve" throughout the whole revised manuscript.

Minor points:

1) *Why not name the phases $Sm_{\{TSB\}}$ and $Sm_{\{SB\}}$ since they are smectics?*

We follow this suggestion and renamed the "twist-splay-bend" and "splay-bend" phases as "twist-splay-bend smectic" and "splay-bend smectic" phases in cases that the presence of density modulations are proven. In other cases we keep the more generic terms "twist-splay-bend" and "splay-bend" phases.

2) *The phrase "chiral symmetry breaking" is commonly used but is also wrong: chirality is an absence of symmetry so it would be more appropriate to call it "achiral symmetry breaking".*

We thank the Referee for the comment that the phrase "chiral symmetry breaking", which is commonly used, is wrong. We rephrased the sentences where we used "chiral symmetry breaking".

This is an important result in the understanding of these new phases that connects molecular shape to phase behavior. The authors should revise the language and discussion but this article should be published after that happens.

REVIEWERS' COMMENTS

Reviewer #1 (Remarks to the Author):

As in my original report, this is a very good paper providing simulation and theoretical evidence for a new liquid crystalline phase. In my view this is a topic of great interest and is likely to inspire new activity.

The authors have answered the queries to my satisfaction and I recommend publication with no further change.

Reviewer #2 (Remarks to the Author):

Dear Editors,

I have read the revised manuscript carefully. The authors have addressed several points of the initial criticism. However, although they recognize in their reply that "the expressions proposed by Meyer and Dozov are based on a continuum theory, which may break down when the variations of the nematic director field are strong as is the case for the TB, SB, and TSB phases reported in our work" they continue to support that the elastic theory is applicable on the molecular scale because "the nematic director field of these phases measured in our simulations are well-fitted by these expressions". In my opinion the physics behind this argument is absent.

In addition, it seems that they are aware of alternative models on the structure, the symmetries and the microscopic origins of the novel nematic phases. I found very disappointing the fact that they do not cite the alternative models in their paper and, more importantly, that they do not attempt to use their excellent simulation results, to enter into the debate on the very origins of this new mode of self organisation, providing in this way a significant advancement in the physics of soft matter. For these reasons I cannot recommend the publication of the paper in a journal with the reputation of NATCOMS.

Reviewer #3 (Remarks to the Author):

The changes made to this paper have improved it from its already excellent state. It is a solid advance in our understanding of these new structural phases.

I would like to address the concerns of referee 2. Phases of matter can be categorized by their symmetries and low energy excitations (Goldstone modes). In this approach a micron-scale colloidal crystal, stabilized by entropic interactions, is the same phase as an covalent, atomic crystal with the same space group. In this way of doing things, the mechanism of interaction, the length scales, and the relaxation times are not part of the identification. (A concrete example would be a smectic-A phase comprised of chiral molecules -- the ordered structure is achiral, despite the fact that there will be an intrinsic chiral response from the constituents. A chiral smectic-A phase would be the twist-grain-boundary phase.)

This is the philosophy adopted in this paper. The phases are named by their symmetries and the authors have made that perfectly clear.

Publish!